# Comment on 'Criterion placement threatens the construct validity of neural measures of consciousness'

Kristian Sandberg[1,2]*, Morten Overgaard[1,3]

[1]Center of Functionally Integrative Neuroscience, Aarhus University and Aarhus University Hospital, Universitetsbyen, Aarhus, Denmark; [2]Neurobiology Research Unit, Copenhagen University Hospital Rigshospitalet, Copenhagen, Denmark; [3]Department of Neurology, Palle Juul-Jensens Boulevard, Aarhus, Denmark

**Abstract** In a recent article Fahrenfort and colleagues demonstrate that auditory and monetary punishment change the criteria that participants use to rate the clarity of perception (Fahrenfort et al., 2025). This leads them to conclude that "subjective measures do not reliably measure the construct they intend to measure" and that their construct validity is threatened. Here, we argue that the main findings had already been demonstrated in other experiments, and the general limitations of subjective measures has been known for decades without the field dismissing their usefulness. We further argue that the experimental manipulations of Fahrenfort and colleagues are so radical that they could be used to invalidate essentially any psychological/cognitive test. In our view, both of the above-mentioned conclusions are thus inappropriate considering the evidence presented, and we believe that the take-home message both before and after the study of Fahrenfort and colleagues is to acknowledge the limitations of subjective reports when they are used.

*For correspondence:
kristian.sandberg@cfin.au.dk

**Competing interest:** The authors declare that no competing interests exist.

## Introduction

*Fahrenfort et al., 2025* examine the effect of auditory and monetary punishment on ratings of visual awareness. As participants alter their response pattern to avoid punishment, the authors conclude that awareness rating scales are vulnerable to criterion effects, which affects neural measurements of consciousness. For example, they report that since the perceptual awareness scale (PAS) was affected by criterion shifts (from a signal detection theory perspective), the construct validity of the measure should be questioned (see their Results section), as should subjective measures in general (see their Discussion section). In this comment, we question the assumptions on which this conclusion is based.

A key worry of Fahrenfort and colleagues is that ratings of "no experience" cannot be taken at face value as such ratings might include instances of very weak experiences. When the criterion is conservative (i.e., a lot of evidence is required to report seeing "a brief glimpse" as opposed to "no experience"), then response accuracy and neural activity related to "no experience" increases. As the authors mention themselves towards the end of the *Introduction*, this criticism is by no means new. And in fact, both key findings – (1) that the amount of estimated subliminal perception varies with criterion changes and (2) that experimental context can change subjective rating response patterns – have been reported in recent articles without prompting us to dismiss subjective measures. As we explain below, the main aspect to keep in mind are the limitations that the use of the measures imposes on the conclusions to be drawn.

## Discussion

### Assumptions of subjective measures

The PAS is a participant-generated awareness rating scale with four steps ranging from "no experience" (called PAS0 and [0] by Fahrenfort and colleagues) to "clear experience" (PAS3) (*Ramsøy and Overgaard, 2004*). In relation to the PAS, Fahrenfort and colleagues report in the *Results* section that the underlying assumption is "that selecting [0] will only occur if trials are 'truly' unseen, so that unconscious processing is not overestimated, as may happen in dichotomous or other types of scales (Overgaard et al., 2006; Overgaard and Sandberg, 2021; Sandberg et al., 2010). The PAS was developed to be able to resolve the inability to externally calibrate subjective content, and as such its ultimate goal seems to be to be impervious to non-perceptual criterion shifts". No such claims were made, however, in the central referenced articles or elsewhere – and working with and developing the PAS, we are in fact often careful to highlight the opposite.

In a number of studies (*Overgaard et al., 2008*; *Ramsøy and Overgaard, 2004*; *Sandberg et al., 2010*), we argue that some "unconscious" effects disappear when using the PAS – potentially because the effects were established using scales with more conservative criteria than the PAS. Nevertheless, this does not mean that any effects found with the PAS should then be considered truly unconscious or that the scale is impervious to criterion shifts/criterion effects. Indeed, we discuss how to identify a "preferred scale" in *Overgaard and Sandberg, 2012* and in *Sandberg et al., 2010*, we specifically compare scales without claiming that any are free from exhaustiveness (i.e., criterion) problems.

The first assumption of Fahrenfort et al. is thus that researchers believe that awareness ratings should be treated as flawless insights into participants' experience. This is a niche position. For example, 32 researchers advise against relying on subjective measures alone to establish evidence of unconscious processing, stating that there is currently no definitive way to determine if above-chance performance is conscious when participants report no experience (*Stockart et al., 2024*, p. 14). In personal communication, Fahrenfort et al. refer to the PAS scale step description as evidence for their claims, but participant instructions and scientific conclusions are clearly not to be equated.

Relatedly, Fahrenfort et al. appear to assume that awareness ratings are only useful if they reflect experience perfectly at all times (otherwise, why would their validity be threatened if we find this not to be the case?). It is trivially true that a non-perfect psychological measure could be clinically relevant for diagnostics and that one with, e.g., a specificity and sensitivity of 0.9 is better than one with 0.8. So, why would we not attempt to identify an optimal subjective measure even if we know it is not perfect? The key issue appears to be to acknowledge the limitation of the measures used.

### Similarity of findings – different conclusions

In one article (*Sandberg et al., 2022*), we specifically discuss the limitations of subjective reports and draw attention to how one may study effects below the (reported) subjective threshold without making assumptions of whether it is truly unconscious processing. Using modelling very similar to that of Fahrenfort and colleagues, we demonstrate that the estimated capacity of subthreshold processing varies as a function of the criterion for reporting awareness, with conservative criteria resulting in larger effects – exactly as Fahrenfort and colleagues find (see, e.g., *Sandberg et al., 2022*, Figure 5B). Nevertheless, we also find that task accuracy above chance is expected to be found for any non-zero criterion and in situations that are unaffected by criterion shifts and regression to the mean effects.

The other main finding of Fahrenfort and colleagues – that report criteria depend on experimental context – has also been established previously. In one article (*Skewes et al., 2021*), we demonstrated this using a false feedback paradigm. When participants received false feedback on accuracy whenever they reported seeing "a weak glimpse" of the stimulus, they changed their response pattern. A similar, yet different, pattern was observed for false confidence ratings feedback, leading us to conclude that awareness and confidence ratings rely on at least partially different processes. That different measures of consciousness lead to different findings, has been reported in several studies (*Lohse and Overgaard, 2019*; *Mazzi et al., 2016*; *Overgaard and Mogensen, 2015*; *Overgaard and Sandberg, 2021*; *Rausch et al., 2015*; *Szczepanowski et al., 2013*). Considering this, it may seem surprising that Fahrenfort and colleagues administered the PAS with multiple adaptations as we explain below.

## Assumptions of instructions

Fahrenfort and colleagues made changes to the PAS – in terms of instructions and administration of punishment of particular reports – which they themselves consider to be so substantial that it may be argued that they did not in fact use the PAS at all (see their *Discussion*). Participants were informed: "*Only press 0 if you are 100% convinced that no square appeared and only press 3 if you are 100% convinced that a square appeared.*" As this encourage participants to report based on confidence, it is not clear what the reports actually represent. For this reason, we have previously attempted to explain how to help participants understand what it involves reporting on experiences avoiding terminology related to confidence (*Sandberg and Overgaard, 2015*).

Fahrenfort and colleagues argue that the changes are unproblematic because there is no such thing as a criterion-free experimental context, and because many dimensions not controlled between experiments will have large effects on criteria. This defence does not take into account that qualitatively different processes (awareness vs. confidence) may be probed.

While participants were encouraged to report honestly using the PAS, they received a secondary set of implicit instructions to avoid certain outcomes reinforced through two types of punishment. The final assumption of Fahrenfort et al. is thus that psychological test responses should be immune to punishment, yet punishment can be used to disrupt the result of essentially any psychological test. Imagine, for example, one group of participants being punished whenever they indicate extraversion in a personality test, and another group being punished for introversion. If punishment is severe enough, the groups would appear to differ greatly on introversion/extraversion. In neuroscientific memory or perception paradigms, we could make the neural correlates increase, decrease or change location if we are creative enough with respect to which responses are punished. But we would not have proven anything other than the effectiveness of punishment.

## Conclusion

Our concern is that the study of Fahrenfort and colleagues primarily adds detail to what was already known about the limitations of subjective measures. While some researchers have perhaps previously concluded too strongly based on subjective measures, we believe that Fahrenfort and colleagues do the same, but in the opposite direction. The study does not attempt to evaluate how large the criterion effects are in natural settings when the PAS is used with recommended instructions and without punishment. The study mainly reestablishes that punishment alters behaviour.

The majority of the criticism raised by Fahrenfort and colleagues can be mitigated simply by acknowledging the limitations of the method as most researchers already do. The conclusions of nearly all studies contrasting more versus less clear experiences (e.g., *Andersen et al., 2016*) are not threatened by the limitations. The magnitude of the differences may vary with criteria, but this is rarely relevant (other than in relation to statistical power) when the aim is to reject a hypothesis of no difference. Another field open to study are effects that occur for all criteria as shown, for example, by modelling (e.g., *Sandberg et al., 2022*). Even current attempts to create theoretical models based on an understanding of consciousness as continuous do not attempt to interpret findings using the PAS as immune to all hypothetical confounds (*Fazekas and Overgaard, 2016*; *Mogensen and Overgaard, 2017*; *Overgaard and Mogensen, 2017*).

## Additional information

### Funding
No external funding was received for this work.

### Author contributions
Kristian Sandberg, Conceptualization, Writing – original draft, Writing – review and editing; Morten Overgaard, Writing – original draft, Writing – review and editing

### Author ORCIDs
Kristian Sandberg ⓘ https://orcid.org/0000-0001-6936-5487
Morten Overgaard ⓘ https://orcid.org/0000-0002-1215-5355

## Decision letter and Author response

Decision letter https://doi.org/10.7554/eLife.106963.sa1

## Data availability

No data was analysed.

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
