## [Decision Letter]

*In the interests of transparency, eLife includes the editorial decision letter. A lightly edited version of the letter sent to the authors after peer review is shown, indicating the most substantive concerns; minor comments are not usually included. Since no essential revisions were requested there is no accompanying author response.*

Thank you for submitting your article 'Comment on 'Criterion placement threatens the construct validity of neural measures of consciousness' to *eLife*. Your article has been reviewed by two peer reviewers, and the evaluation has been overseen by a Reviewing Editor (Ming Meng) and a Senior Editor (Joshua Gold).

Reviewer Comments

*Reviewer #1:*

This commentary by Sandberg and Overgaard disagrees with Fahrenfort et al.’s (2025) main conclusion. Fahrenfort et al. say that subjective measures (like the PAS scale) are invalid because they are affected by how people set their decision criteria ("criterion effects"). Sandberg and Overgaard contend that: (1) the limitations of subjective reports are already well-established in the field; (2) Fahrenfort et al.'s experimental manipulations (punishment, modified instructions) are too extreme to generalize; and (3) the study fails to contextualize its findings within existing literature. The rebuttal is well-supported by prior work but could more explicitly propose solutions for mitigating criterion effects.

Substantive concerns

1. The Epistemological Dilemma of "True" Unconsciousness:

Both papers grapple with the fundamental impossibility of directly verifying subjective states. Fahrenfort et al. implicitly assume that a "truly unseen" state exists and can be reliably isolated with a perfect measure. Sandberg & Overgaard correctly note that establishing this "ground truth" is impossible.

2. Conflation of Measurement Validity with Construct Existence:

Fahrenfort et al.'s conclusion that criterion shifts "threaten construct validity" risks conflating the measurement of a construct with the existence of the construct itself. Their manipulations show that reports of awareness are malleable, but this does not invalidate the construct of perceptual consciousness or the neural correlates studied. Sandberg & Overgaard touch on this by noting neural correlates of graded awareness remain valid.

3. Lack of Engagement with the Purpose of Graded Scales:

Fahrenfort et al.'s focus on the malleability of the "no experience" criterion overlooks the primary purpose of graded scales like the PAS: capturing gradations in conscious clarity. These scales are designed to distinguish "clear" from "vague" experiences within reportable consciousness, enabling the study of neural correlates of graded awareness. While criterion shifts affect category boundaries, they may have less impact on neural distinctions between extremes (e.g., PAS3 vs. PAS1-2).

*Reviewer #2:*

The critique questions the originality of Fahrenfort et al.'s paper, arguing that the main findings were already well-known and that there is a risk of misinterpretation in the experimental instructions of Experiment 2, rendering the data insufficient to support the conclusions. However, the listed main findings in the text seem to have been misunderstood. The key findings mentioned in the introduction are not the main conclusions of the study but rather assumptions for data simulation and neural decoding analysis. The critique's hypotheses in the discussion section also do not align with the original text. Criticisms based on misinterpreted findings and assumptions are unfounded. It is recommended that the authors analyze and explain how the misinterpretation of key findings and assumptions occurred. If there are issues in the original text leading to misinterpretation, clear revision suggestions should be proposed to reduce ambiguity in the article and minimize the risk of misinterpretation.